# Effect-Directed Profiling of Powdered Tea Extracts for Catechins, Theaflavins, Flavonols and Caffeine

**DOI:** 10.3390/antiox10010117

**Published:** 2021-01-15

**Authors:** Gertrud E. Morlock, Julia Heil, Antonio M. Inarejos-Garcia, Jens Maeder

**Affiliations:** 1Institute of Nutritional Science, and TransMIT Center for Effect-Directed Analysis, Justus Liebig University Giessen, Heinrich Buff Ring 26–32, 35392 Giessen, Germany; Julia.Heil@ernaehrung.uni-giessen.de; 2Department of Functional Extracts, ADM Wild Valencia, 46740 Carcaixent, Spain; Antonio.Inarejos@adm.com; 3Department of Science & Technology, ADM Wild Europe, 13597 Berlin, Germany; Jens.Maeder@adm.com

**Keywords:** functional food, health food, adulteration, falsification, food safety, bioassay, enzyme assay, flavonoid, high-performance thin-layer chromatography, *Camelia sinensis*

## Abstract

The antioxidative activity of *Camelia sinensis* tea and especially powdered tea extracts on the market, among others used as added value in functional foods, can considerably vary due to not only natural variance, but also adulteration and falsification. Thus, an effect-directed profiling was developed to prove the functional effects or health-promoting claims. It took 3–12 min per sample, depending on the assay incubation time, for 21 separations in parallel. Used as a fast product quality control, it can detect known and unknown bioactive compounds. Twenty tea extracts and a reference mixture of 11-bioactive compounds were investigated in parallel under the same chromatographic conditions by a newly developed reversed phase high-performance thin-layer chromatographic method. In eight planar on-surface assays, effect-directed tea profiles were revealed. Catechins and theaflavins turned out to be not only highly active, but also multi-potent compounds, able to act in a broad range of metabolic pathways. The flavan-3-ols acted as radical scavengers (DPPH^∙^ assay), antibacterials against Gram-positive *Bacillus subtilis* bacteria, and inhibitors of tyrosinase, α-glucosidase, β-glucosidase, and acetylcholinesterase. Further effects against Gram-negative *Aliivibrio fischeri* bacteria and β-glucuronidase were assigned to other components in the powdered tea extracts. According to their specifications, the activity responses of the powdered tea extracts were higher than in mere leaf extracts of green, white and black tea. The multi-imaging and effect-directed profiling was not only able to identify known functional food ingredients, but also to detect unknown bioactive compounds (including bioactive contaminants, residues or adulterations).

## 1. Introduction

Tea (*Camelia sinensis*) is the most consumed drink after water, with an increasing tendency [1]. Tea drinking is considered healthy due to its inherent antioxidative polyphenolic compounds, e.g., flavan-3-ols like catechins and theaflavins as well as flavonols [2]. As a trend, functional tea extracts are more and more added to industrial food products to increase the health benefit for consumers (nutraceuticals, functional food and health food) [3,4]. Most tea and tea extracts are imported, especially from China, where half of the tea plantations worldwide are located [5]. However, the quality of tea extracts and tea products on the market can be quite different. This is especially true for powdery products that are susceptible to falsification, counterfeiting or imitation, as it is more difficult to prove authenticity after comminution [6,7,8,9]. In addition, blending of tea leaves or powders is commonly used for processing to guarantee a constant tea quality and compensate for deviations caused by cultivation and environment [10,11]. However, this blending practice could also easily be misused. Further doubts exist regarding the extraction solvent used to produce the powdered tea extracts. Often these are specified to be aqueous extracts, but the volatile profile shows traces of other solvents [12]. Therefore, the current control of powdered tea extract products needs to be investigated. In particular, bioanalytical screening tools are needed to check these functional food products.

The catechins that are inherent in green tea have the potential to impact a variety of human diseases [13,14,15], but also in black tea, the theaflavins are reported to have beneficial properties [16,17]. The latter reddish–orange theaflavin pigments are dimers with a benzotropolone moiety formed from catechins by enzymatic oxidation during the fermentation of tea. These and further oxidation products like theasinensins and (epi)theaflagallins [18] can react further during processing into more complex brown–red thearubigins of dimeric to polymeric acidic structures [19,20]. The strong antimicrobial, antiviral, antifungal, anticarcinogenic [21,22,23,24] and antioxidant [25,26] effects of tea and tea extracts are mainly attributed to the catechins, but also partially to the theaflavins. Thus, tea consumption can reduce the risk of cardiovascular diseases, protect against diabetes [27] and Alzheimer’s disease [28,29], reduce skin diseases, help with digestive disorders and respiratory diseases [18]. In order to substantiate postulated functional, health-promoting qualities of food products containing powdered tea extracts, e.g., associated with an increased intake of catechins and theaflavins, further evidence-based research is necessary.

The present analytical methods of product control are challenged by the growing product segment of functional or healthy food on the one hand, and generally increasing cases of adulteration and falsification on the other hand. Methods are often tailored with regard to extraction, separation and detection to the analysis of target compounds, and are blind [30] for detecting unknown adulterations or other unauthorized additions along the global production chain. Additionally, contaminants [31] or residues [32] in tea may impact the health of consumers. For example, high-performance liquid chromatography [33,34,35], high-performance thin-layer chromatography (HPTLC) [36,37,38] and gas chromatography [39,40] are used to analyze the polyphenolic compounds and essential oils of tea, respectively. Theaflavin pigments from black tea were mostly analyzed by HPLC, but not by HPTLC. As the state of the art, five main catechins of green tea have been separated at once on cellulose TLC plates so far [41]. Except for one method on green tea (extract) [42], most HPTLC methods on tea were focused on individual target catechins, but not on characteristic profiles of the functional powdered products obtained from different tea types. Therefore, the question arose as to whether other tools, particularly bioanalytical tools, could be of use in the analysis of functional or health foods.

In this study, a fast effect-directed, multi-imaging HPTLC profiling was developed for tea and especially powdered tea extracts. In contrast to current methods, it exploits biological and enzymatic detections to directly point to bioactive (functional) compounds. This is highly important, as both known and unknown bioactive compounds are detectable, that may affect the health of consumers. It was intended to prove the authenticity (characteristic profiles) and involved effects deriving from individual ingredients (bioactivity profiles) on the one hand, and to discover product defects and undesirable components (multi-imaging profiles) on the other hand. Seventeen commercially available powdered tea extracts on the market were compared with certified black, white and green tea leaves from China and a reference mixture of 11 bioactive compounds.

## 2. Materials and Methods

### 2.1. Chemicals

All salts (per analysis) were waterfree. Other purity grades were listed, if available. All solvents were of chromatography grade. Methanol was from VWR International, Darmstadt, Germany, and acetonitrile from Honeywell Specialty Chemicals, Seelze, Germany. (+)-Catechin (C, 98%), (-)-gallocatechin (GC, 89%), (-)-catechin-3-gallat (CG, 96%), (-)-epicatechin (EC, 100%), (-)-epicatechin-3-gallat (ECG, 98%), (-)-epigallocatechin (EGC, 95%), (-)-epigallocatechin-3-gallat (EGCG, 96%), rutin (R, 90%), and theaflavin (T, 94%) were obtained from by PhytoLab, Vestenbergsgreuth, Germany. Quercetin (Q, 95%), caffeine (Caf, ≤100%, waterfree), acarbose (for pharm.), α-glucosidase from *Saccharomyces cerevisiae*, β-glucuronidase from *Escherichia coli*, acetylcholinesterase (AChE) from *Electrophorus electricus*, tyrosinase from mushroom, Müller–Hinton broth (for microbio.), peptone from casein (tryptone, for microbio.), diammoniumhydrogen phosphate (99%), sodium acetate, di-sodium hydrogen phosphate, monopotassium phosphate, magnesium sulfate heptahydrate, sodium chloride, rivastigmine tartrate (≥98%) and imidazol (≥99.5%) were delivered by Sigma–Aldrich, Steinheim, Germany. β-Glucosidase from almonds were provided by ABCR, Karlsruhe, Germany. Tetracycline hydrochloride (reagent grade) was purchased from by Serva Electrophoresis, Heidelberg, Germany. D-Saccharolactone was obtained from Santa Cruz Biotechnology, Dallas, TX, USA. Fast Blue B salt (95%) was from MP Biomedicals, Eschwege, Germany. 5-Bromo-4-chloro-3-indolyl β-D-glucuronide sodium salt (X-glucuronide, ≥98%) was obtained from Carbosynth, Compton–Berkshire, UK. 2,2-Diphenyl-1-picrylhydrazyl (DPPH^∙^, 95%) was delivered by Alfa Aesar, Schwerte, Germany. Bovine serum albumin (BSA, fraction V, ≥98%), 3-(4,5-dimethylthiazolyl-2)-2,5-diphenyl-2H-tetrazolium bromide (MTT, ≥98%), 4-methylumbelliferyl-α-D-glucopyranoside, koji acid (>98%), gallic acid (≥98%), indoxyl acetate, natural product reagent (Naturstoffreagenz A, ≥98%), 3-[(30cholamidopropyl) dimethylammonium]-1-propanesulfonate (CHAPS, ≥98%), tris(hydroxymethyl)aminomethane (Tris, ≥99.9%), dipotassium hydrogen phosphate (≥99%), potassium dihydrogen phosphate (99%), disodium hydrogen phosphate (≥99%), sodium dihydrogen phosphate monohydrate (≥98%), sodium hydroxide (≥98%), glycerol (Rotipuran, 86%), polyethylene glycol (PEG) 8000 (Ph. Eur.), anisaldehyde (4-methoxybenzaldehyd), sulfuric acid and glacial acetic acid (both >98%) were from Roth, Karlsruhe, Germany. Yeast extract powder (for microbiol.) was purchased from Th. Geyer, Renningen, Germany. (2S)-2-Amino-3-(3,4-dihydroxyphenyl)propanoic acid (levodopa, 99%) was ordered by J&K Chemicals Chandigarh, India. Dulbecco´s phosphate-buffered saline (DPBS) salt from Biochrom GmbH, Berlin. The medium for the Gram-negative, bioluminescent marine *Aliivibrio fischeri* bacteria (DSM–7151, German Collection of Microorganisms and Cell Cultures, Berlin, Germany) is listed elsewhere [43]. Gram-positive soil bacteria *Bacillus subtilis* (Bundesgesundheitsamt, BGA) spore suspension, citric acid monohydrate as well as water-wettable (W), reversed phase (RP) and acid-stable fluorescence indicator containing (F_254_s) LiChrospher^®^ HPTLC plates silica gel 60 RP–18 WF_254_s (20 × 10 cm, spherical particles) and HPTLC plates silica gel 60 RP–18 WF_254_s (only available as 10 × 10 cm format, irregular particles) were provided by Merck, Darmstadt, Germany. Bidistilled water was produced (Heraeus Destamat Bi–18E, Thermo Fisher Scientific, Dreieich, Germany).

### 2.2. Stock Solutions, Standard Mixture and Sample Extracts

Stock solutions (2.2 µg/µL each of Caf, C, GC, CG, EC, ECG, EGC, EGCG and T) in acetonitrile (in methanol for R and Q) were ultrasonicated for 5 min. A standard mixture (1.1 mL) was prepared by mixing together 100 µL each of the 10 flavonoids and Caf in a sampler vial (0.2 µg/µL each). As stock solutions, 20 *Camellia sinensis* samples of different suppliers and standardizations (ID 1–20, Appendix A), i.e., 3 certified dried leaves of green, white and black tea (grinded using mortar and pistil, then sieved to 500–µm particle size) and 17 commercially available powdered extracts, were extracted with methanol (100 µg/µL each), vortexed (30 s), ultrasonicated (15 min) and centrifuged (17,000× *g*, 10 min). After 1:10 dilution with methanol (10 µg/µL each), each was transferred to a sampler vial. All solutions were stored at −20 °C.

### 2.3. Plate Prewashing and Remarks

A set of HPTLC plates was developed with methanol–water, 4:1 (*V/V*) up to the upper plate edge in the Simultan Separating Chamber (biostep, Burkhardtsdorf, Germany), dried in an oven at 110 °C for 20 min, covered by a clean counter glass plate (placed on top of the stacked plates) and wrapped in aluminum foil for storage in a desiccator. As standards of high purity are expensive, the application parameters were selected so that less microliter volume was lost for the automated syringe operation (filling vacuum time 1 s; rinsing vacuum time 6 s; rinsing/filling cycles 1; return unused sample into vial; Automatic TLC Sampler ATS4, CAMAG, Muttenz, Switzerland). Plate drying was always performed in a stream of cold air (hair dryer or Automatic Developing Chamber 2, ADC 2, CAMAG), immediately after application (0.5 min) and development (5 min). The relative humidity of the ambient air was ca. 40 ± 5% during the developments. If required, this humidity of the plate can be adjusted using a saturated sodium acetate solution. All instrument operation and acquired data were processed with the software visionCATS (version 3.0, CAMAG).

### 2.4. RP–HPTLC Method

The 21 different solutions were sprayed on the HPTLC plate as 21 tracks as follows (ATS4, CAMAG): band length 7 mm, track distance 8.5 mm, distance from lower edge 8 mm and from left edge 15 mm, dosage speed 150 nL/s, application volumes 1 µL/band for sample extracts and 3 µL/band for the standard mixture (0.6 µg/band each), if not stated otherwise. The development (20 × 10 cm Twin Trough Chamber, biostep, or ADC 2, CAMAG) was performed with 7 mL acetonitrile–water–citric acid (3 mL + 6 mL + 15 mg) on the LiChrospher^®^ HPTLC plate silica gel 60 RP–18 WF_254_s. The ratio was changed to 1.8 mL + 6 mL + 23 mg for development (respective 10 × 10 cm chamber) on the HPTLC plate silica gel 60 RP–18 WF_254_s. Both separations took ca. 25 min up to a developing distance of 8 cm, measured from the lower plate edge. Documentation (TLC Visualizer, CAMAG) was performed at 254 nm (UV), 366 nm (FLD) and under white light illumination (Vis). Two different derivatization reagents were piezoelectrically sprayed (3 mL each, spraying level 5, Derivatizer, CAMAG) as reagent sequence, one after the other on the same plate: first 0.5% aqueous Fast Blue B salt reagent (prepared freshly, red nozzle), followed by plate heating (100 °C, 3 min; TLC Plate Heater, CAMAG) and documentation/densitometry at Vis, then 1% methanolic natural product reagent (green nozzle) and documentation/densitometry at FLD. For method development, anisaldehyde sulfuric acid reagent (4-methoxybenzaldehyd–sulfuric acid–glacial acetic acid–methanol 0.05:1:1:9, *V*/*V*/*V*/*V*, blue nozzle, level 3; followed by heating at 110 °C for 2 min) was used as this reagent detected all 10 flavonoids at one go. Densitometry was performed at UV 275 nm (absorbance measurement of caffeine, but also all other compounds; deuterium lamp), after derivatization with the Fast Blue B salt reagent at 546 nm (absorbance measurement of flavan–3-ols; mercury lamp) and after derivatization with the natural product reagent at FLD 366/>400 nm (fluorescence measurement of flavonols; mercury lamp).

### 2.5. Effect–Directed Profiling

Eight chromatograms were prepared analogously. The intrinsic pH value of the LiChrospher^®^ HPTLC plate silica gel 60 RP–18 WF_254_s was measured to be pH 4.0, and after the acidic development even pH 3.1. Hence, the dried chromatogram was neutralized with 2.8 mL sodium hydrogen carbonate buffer (2.5 g/100 mL, pH 8) by piezoelectric spraying (yellow nozzle, level 6) and dried for 4 min. For each assay, a respective positive control (PC) was applied bandwise at a poorly, sufficiently and well detectable amount on the upper plate edge. Then, as mentioned in the respective assay, only a few milliliters of the assay solutions were piezoelectrically sprayed (if not stated otherwise, blue nozzle, level 6, Derivatizer, CAMAG). For incubation, the chromatogram was placed horizontally in a polypropylene box (27 × 16 × 10 cm, KIS, ABM, Wolframs–Eschenbach, Germany), which was pre-moistened with 35 mL water spread on filter papers aligned on walls and bottom at room temperature for 30 min. After final plate drying (3 min), FLD/Vis/bioluminescence images were documented, whereby the *B. subtilis* and DPPH^∙^ images were captured again after one day, as their response signals increased.

(1)DPPH^∙^ assay: 4 mL 0.04% methanolic DPPH^∙^ solution was sprayed (green nozzle, level 4). Yellow bands on a purple background were generated instantly. The PC was gallic acid (0.5, 1.3 and 2 µL/band, 0.1 mg/mL in methanol).(2)Gram-negative *A. fischeri* bioassay: 150 µL bacterial cryostock were incubated in 20 mL medium according to DIN EN ISO 11348–1 [43] in a 100 mL culture flask at 75 rpm and room temperature for 18–24 h. By shaking the culture flask in a dark room, the green–blue bioluminescence of the bacteria was visually proven to be ready for use, and 4 mL bacterial suspension were sprayed on the plate. The settling down of the vapor was interrupted to transfer the still humid plate to the BioLuminizer cabinet (CAMAG). Fifteen images were recorded over 45 min (exposure time of 60 s, trigger interval 3.0 min). Antibiotics were detected as dark or brightened bands on the instantly bioluminescent plate background. The PC was Caf (0.5, 1.5 and 3 µL/band, 1 mg/mL in methanol).(3)Gram-positive *B. subtilis* bioassay, as exception recommended on 10 × 10 cm HPTLC plate silica gel 60 RP–18 WF_254_s: 3.5 mL bacteria suspension (100 µL bacterial cryostock per 20 mL 2.3% Müller–Hinton broth, optical density of ca. 0.8 at 600 nm [44]) were sprayed (red nozzle), followed by incubation at 37 °C for 2 h. For generation of the colorless (white) antibacterial bands on a purple background, the plate was sprayed with 500 µL 0.2% DPBS-buffered MTT solution and incubated at 37 °C for 45 min, followed by drying (50 °C, 5 min, Plate Heater, CAMAG). The PC was tetracycline (0.5, 1.5 and 3 µL/band, 0.005 mg/mL in ethanol). The application of this bioassay on the LiChrospher^®^ HPTLC plate silica gel 60 RP–18 WF_254_s required an additional binder hardening by heating the plate at 140 °C for 20 min, a two-fold neutralization of the more acidic layer using a pH 12 buffer (citric acid 6 g/L, di-sodium hydrogen phosphate 10 g/L), a longer (overnight) incubation and higher sample amounts to be applied (as not so sensitive in the detection on this layer).(4)α-Glucosidase inhibition assay: 2 mL substrate solution (12 mg 4-methylumbelliferyl-α-D-glucopyranoside dissolved in 0.2 mL dimethyl sulfoxide and diluted with 9 mL ethanol and 1 mL 10 mM sodium chloride solution) were sprayed first (green nozzle, level 5), then after drying (2 min), 2.5 mL α-glucosidase solution (10 U/mL in sodium acetate buffer, pH 7.5), followed by incubation (37 °C, 90 min) and drying (3 min). To obtain the most intense 4-methylumbelliferyl-blue fluorescent background, the plate was made alkaline by placing it (15 min) in a dry chamber with the counter trough filled with 10 mL 25% ammonia solution. As contrast, dark inhibition bands absorbing at FLD 366 nm were revealed. The PC was acarbose (1, 3 and 6 µL/band, 3 mg/mL in ethanol).(5)β-Glucosidase inhibition assay: same as before, but the substrate was 4-methylumbelliferyl-β-D-glucopyranoside, the β-glucosidase solution was 1000 U/mL and the incubation took 90 min. The PC was imidazole (2, 5 and 8 µL/band, 1 mg/mL in ethanol).(6)Tyrosinase inhibition assay: 2 mL were sprayed each of substrate solution (4.5 mg/mL levodopa in phosphate buffer of 0.14% dipotassium phosphate and 0.16% disodium phosphate, 20 mM, pH 6.8, plus 2.5 mg CHAPS and 7.5 mg PEG 8000), and after drying (1 min), tyrosinase solution (400 U/mL in phosphate buffer), followed by incubation at room temperature for 15–20 min to reveal the colorless (white) inhibition bands on a grey background. The PC was kojic acid (1, 3 and 6 µL/band, 0.1 mg/mL in ethanol).(7)AChE inhibition assay: The plate was sprayed (green nozzle) with 1.3 mL substrate solution (1 mg/mL indoxyl acetate in ethanol) and then 3 mL AChE solution (6.66 U/mL in Tris–HCl buffer plus 1 mg BSA), followed by incubation at 37 °C for 25 min. Visible indigo-blue inhibition bands were revealed, which were more sensitively detected as absorbing dark bands on the indigo-blue fluorescent plate background at FLD 366 nm. The PC was rivastigmine (2, 4 and 8 µL/band, 0.1 mg/mL in methanol).(8)β-Glucuronidase inhibition assay: 2.0 mL of β-glucuronidase solution (50 U/mL in potassium phosphate buffer, 0.1 M, pH 7.0) were sprayed (yellow nozzle), followed by incubation at 37 °C for 15 min [45]. For generation of the colorless (white) inhibition bands on an indigo-blue colored background, the plate was sprayed (red nozzle) with 1.5 mL 5-bromo-4-chloro-3-indolyl-β-D-glucuronide solution (2 mg/mL in water) and incubated at 37 °C for 1 h. The PC was D–saccharolactone (0.5, 1.5 and 3 µL/band, 0.1 mg/mL in water).

## 3. Results and Discussion

### 3.1. Setup of the Profiling

On the one hand, the number of plant powders (botanicals) specified to be rich in healthy compounds and used as added value in functional foods is steadily increasing [1,3]. On the other hand, no cost-efficient bioanalytical screening tools are used in product control to prove these postulated functionalities. Hence, a bioactivity profiling was developed on RP–HPTLC plates to detect in such products not only known, but also unknown bioactive compounds that may affect the health of consumers. Therefore, planar chromatographic sample separations were combined in situ with biological and enzymatic assays. The use of a wettable apolar layer material was precondition for the subsequent application of the very polar assay solutions or bioassay suspensions. Eleven bioactive compounds [2,15,31] were selected as references for the tea profiling, i.e., 10 flavonoids and the methylxanthine alkaloid Caf. Among the 10 flavonoids were 8 flavan-3-ols (i.e., the catechins C, GC, CG, EC, ECG, EGC, EGCG and the main theaflavin T) as well as exemplarily two flavonols (R and Q). This 11-bioactive-compound mixture, of which bioactivity responses were expected in the planar assays, was used for the method development.

For verification and application of the developed profiling, 17 functional powdered tea extracts on the market (Appendix A) were investigated with regard to bioactivity (functional) responses and compared with the 11-bioactive-compound mixture and with certified black, white and green tea leaves from China, which were employed as quality reference leaves. Therefore, the 21 separations were performed in parallel on the same plate and simultaneously subjected to each effect-directed assay (EDA). Eight different assays were selected and applied, which directly pointed to individual radical scavenging compounds (antioxidants), antibacterials against Gram-negative and Gram-positive bacteria as well as enzyme inhibitors against α-glucosidase, β-glucosidase, tyrosinase, AChE and β-glucuronidase. A plate prewashing was recommended [46,47] to obtain clear signal responses in each assay. Instead of the commonly used automated immersion of the plate, the assay solutions and suspensions were cost-efficiently applied by piezoelectric spraying, which required only few milliliters and avoided a zone shift or blurring of the main flavonoids in the tea samples when the plate was pulled out [48]. The proper performance of each assay was proven and documented by a respective PC, located above the solvent front in the upper plate edge. The successful color formation of the plate background was considered as negative control. A blank of the extraction solvent was also applied and proven to generate no response in the different assays.

### 3.2. Method Development

Attempts to use mobile phase systems for tea analysis in reported HPTLC methods [37,38,41,42] led to coelution of the flavonoids. So far, 11 bioactive compounds have not been shown to be separated at one go in HPTLC. Therefore, the method development was done from scratch. On polar silica gel and modified middle-polar layers (ca. 50 different mobile phases tested, data not shown), the resolution of the 11 bioactive compounds was not as good as on RP–HPTLC plates. Nevertheless, the method development on RP–phases took another 32 separations (Appendix A). The selection and use of wettable apolar layer materials (Appendix A) was an important criterion and pre-condition for proper penetration of the subsequently applied polar assays into the adsorbent. A separation of all 11 bioactive compounds was finally achieved best using a mixture of acetonitrile, water and citric acid on the irregularly particulate HPTLC plate silica gel 60 RP–18 WF_254_s (Figure 1).

This developed mobile phase system was similar to the one (although containing formic acid) applied for *Salix* samples, which was found later in literature [49] and tested in comparison (Appendix A). However, the own experience in mobile phase development clearly showed that even small variations in the solvent ratio had a high impact on the resolution of the 11 compounds. In addition, the given amount of T did also influence the resolution to the adjacent Q. With increasing T amounts, its *hR*_F_ value increased. For T amounts higher than 0.2 µg/band, it was observed that T coeluted with Q, and for even higher amounts, both changed their elution order (Appendix A). Nevertheless, Q and T can be detected separately because HPTLC allows a post-chromatographic selective derivatization via the natural product reagent (Figure 1b,c). This was also evident in the respective FLD densitogram at 366/>400 nm obtained after derivatization (Figure 2d). At higher amounts per band, meaning higher sample application volumes, all 11 compounds were detectable via their absorbance in the chromatogram at UV 254 nm (Figure 1a). This was also observed in the UV densitogram at 275 nm (Figure 2b).

Moreover, a more sensitive and selective detection of the bioactive compounds was thought to be advantageous, in particular for matrix-rich tea products. A detailed investigation of derivatization reagents followed. The chloramine-T, ferric chloride, ferric chloride-iodine-tartaric acid, vanillin sulfuric acid, anisaldehyde sulfuric acid, natural product reagent (plus PEG), Fast Blue B salt and *p*-dimethylaminocinnam-aldehyde reagents were applied solely, and also as reagent sequence in different combinations and orders (data not shown). The Fast Blue B salt reagent was sprayed on and revealed most sensitively the 8 flavan-3-ols as purple-red bands (Figure 1b and Appendix A). The natural product reagent was selectively detecting both flavonols (Figure 1c). The Q was observed as yellow and R as orange fluorescent bands. In addition, the three gallates (EGCG, ECG and CG) turned to be blue fluorescent. The commonly applied PEG (for fluorescence stabilization or enhancement) had no additional advantage (Figure 1c and Appendix A). Only the fluorescence color of Q tuned from yellow to orange. Among the investigated combinations, the most selective and sensitive reagent sequence was as follows (Figure 1d): (1) the Fast Blue B salt reagent was sprayed on (for Vis detection of flavan-3-ols), then on the same plate, (2) the natural product reagent was sprayed on (for FLD detection of flavonols). Initially, for fast visual evaluation of the zone resolutions in the tested mobile phases during method development, the anisaldehyde sulfuric acid reagent was used (Appendix A). This reagent had the advantage that all 10 flavonoids were detectable at FLD 366 nm after a moderate, short heating. However as mentioned, a more sensitive and selective detection was achieved using the reagent sequence of Fast Blue B salt reagent and then natural product reagent.

In summary, the selected 11 bioactive compounds were recommended to be measured for quantification as follows: the UV absorbance of caffeine at 275 nm, the Vis absorbance of flavan-3-ols at 546 nm after derivatization with the Fast Blue B salt reagent, and the fluorescence of flavonols at 366/>400 nm after derivatization with the natural product reagent (Figure 2). It was observed that the UV/FLD signal responses slightly increased in general (also for other compounds present in the extracts) after the application of the buffer solution, which was used for layer neutralization before the assay application (Appendix A). Hence, the UV 275 nm measurement was recommended to be performed after the neutralization step to better detect the minor compounds. Reported UV response increases of catechins over time [49] were not observed on the buffered RP plate at UV 275 nm (Appendix A). First calibrations (with mean standard deviations of 1.5%, *n* = 6) and precisions of repeated sample measurements (mean precision of 4% over the 11 compounds, *n* = 4) were satisfying, but are the focus of a separate quantitative method comparison study.

The separation was also performed on the LiChrospher^®^ HPTLC plate silica gel 60 RP–18 WF_254_s, made of spherical particles and packed more homogeneously, which could improve the separation. However, the separation was not better, but comparably good (Figure 3). Astonishingly, T and Q changed the position, but were still resolved. The whole analyte range was shifted to higher *hR*_F_ values by increasing the polarity of the solvent system (3 mL + 6 mL + 15 mg), which was found to be advantageous for tea samples rich in matrix or theaflavins. Both separations either on RP–18 W plates with irregular or spherical particles took about 25 min up to a developing distance of 8 cm from lower plate edge. On the 20 × 10 cm format of the LiChrospher^®^ HPTLC plate silica gel 60 RP–18 WF_254_s, the physicochemical profiling could be performed simultaneously for all 21 samples (Figure 3a). The separation was calculated to be about 1 min per sample, and the whole physicochemical profiling 3 min per sample.

In the flavan-3-ol profiles obtained by the Fast Blue B salt reagent (Figure 3b), the theaflavins (below *hR*_F_ 20) were clearly visible in the black tea extracts, but not in green tea extracts. Flavonols were present in both but not all black and green tea extracts, as evident in the respective profiles obtained by the natural product reagent (Figure 3c, yellow or orange fluorescent zones). Tea extract IDs 2 and 19 (Appendix A) are the commercially available extracts of the respective certified tea leaf material (IDs 1 and 20), and thus were expected to have a similar pattern. Tea extract IDs 5 and 11 list a lower polyphenol content in the specifications (20% and 15% polyphenols, respectively), which is in accordance with the less intense pattern of the bioactive compounds, if compared to other extracts. The HPTLC–UV/Vis/FLD profiling clearly showed the difference in the individual contents of bioactive compounds in tea extracts on the market. Additionally, similarities were observed, as for sample IDs 8 and 9 (95% and 98% polyphenols specified, respectively). As evident, the given specifications are very rudimental and quite different in the information, i.e., based on the EGCG, T, Caf, catechins or polyphenols contents (Appendix A). Hence, the characteristic HPTLC tea patterns deliver fast additional (more comprehensive) information. Any deficiency, falsification or adulteration based on the non-volatile part would also be detectable by this multi-imaging owed to the five different detection principles used on the same plate (UV/Vis/FLD and two different derivatizations applied as reagent sequence).

### 3.3. Development of the Effect-Directed Profiling

Eight LiChrospher^®^ chromatograms were prepared analogously and subjected to the different assays. The effect-directed HPTLC–EDA–UV/Vis/FLD/bioluminescence profiling of the 20 tea samples directly pointed to bioactive (functional) compounds therein (Figure 4). For the enzymatic assays against α-glucosidase, β-glucosidase, AChE and β-glucuronidase, 10 µg/band of each tea sample and 0.6 µg/band of each bioactive compounds were applied. When a higher amount of each extract (15 µg/band) was applied, the zones were more pronounced as observed in the tyrosinase autogram, or vice versa for lower amounts applied, less pronounced as given for the 11-compound mixture (0.2 µg/band). However, to avoid response overload, the tea extract amounts were reduced to 10, 7 and 2 µg/band for the Gram-negative *A. fischeri*, Gram-positive *B. subtilis* and radical scavenging assay, respectively. The *B. subtilis* bioassay was recommended to be performed on the HPTLC plate silica gel 60 RP–18 WF_254_s using acetonitrile–water–citric acid (1.8 mL + 6 mL + 23 mg), as the more acidic LiChrospher^®^ plate did not lead to a satisfying response at the same sample amounts, even not after a 24–h incubation. The plate buffering with the sodium hydrogen carbonate buffer (pH 8) seemed to be not sufficient, as evident in the initially poor background color formation (Appendix A). A repeated immersion in this buffer led to furrows (Appendix A), and thus, a pH-12 buffer was applied by a two-fold immersion. However, the layer fixation on the glass carrier was challenged by the multiple (>4) applications of mostly aqueous solutions used for prewashing, neutralization and assay media. Thus, a heating of the plate layer (hardening of the binder) was highly recommended to avoid flaking off the layer (Appendix A) or formation of streaks (Appendix A). Finally, a response was obtained also on this layer, however, the response was not as sensitive. Even when 20 µg of each sample (instead of 7 µg, Figure 4) was applied and the incubation was performed overnight, still an about 20-fold weaker response was observed. Obviously, one or more ingredients used in the production of the LiChrospher^®^ layer are not compatible with the *B. subtilis* bioassay application. Out of the 8 assays, only the *B. subtilis* bioassay was impaired on the LiChrospher^®^ layer. The layer influence on the bioassay application and on its results is focus of a separate study.

The chromogenic Fast Blue B salt substrate commonly used for the AChE, α-glucosidase and β-glucosidase inhibition assays was changed, as the Fast Blue B salt itself reacted with the bioactive compounds. Two different options were demonstrated to successfully solve such otherwise false positive reactions (Figure 4). On the one hand, 4-methylumbelliferyl-α-D-glucopyranoside and 4-methylumbelliferyl-β-D-glucopyrano-side were used as substrates for the α-glucosidase and β-glucosidase, respectively, producing dark inhibition bands absorbing on a 4-methylumbelliferyl-blue fluorescent background at FLD 366 nm. On the other hand, indoxyl acetate was applied as substrate for the AChE, generating dark inhibition bands absorbing at FLD 366 nm on an indigo-blue fluorescent background, or less sensitively detected as indigo-blue bands at Vis. However, this substrate converted theaflavins (initially not colored on the plate for the used low sample amounts per zone, Figure 3a) into yellow (marked*) instead of indigo-blue zones in the Vis image. By this, another, so far unknown theaflavin representative was detected in the tea sample IDs 3, 8, 9, 13 and 15–18, coeluting with EGCG. The comparative evaluation of both FLD/Vis images obtained by the AChE inhibition assay was helpful for result interpretation. The coeluting zone needs further clarification via online elution into a high-resolution mass spectrometer, made in a separate study.

### 3.4. Results of the Effect-Directed Profiling

The effect-directed profiling was performed for 21 samples in parallel, and it was calculated to take 3–12 min per sample, depending on the incubation time of the planar assay. This was found to be a rapid bioanalytical screening, suited for routine control of functional food products. In all eight assays, multiple effects were detected in most samples. The catechins and theaflavins were proven to be multi-potent compounds, able to act in different metabolic pathways. The individual theaflavins and catechins (and few further compounds above GC) were detected to be radical scavenging compounds (DPPH^∙^ assay), antibacterials against Gram-positive *B. subtilis* bacteria, and inhibitors of tyrosinase, α-glucosidase and AChE. These findings were in accordance with literature reporting strong antioxidant [25,26], antibacterial [21,22,23,24] activities as well as protective effects against diabetes [27] and Alzheimer’s disease [28,29]. All 8 flavan-3-ols showed almost comparable radical scavenging, α-glucosidase and AChE inhibiting effects, whereas the antibiotic activity against Gram-positive bacteria was pronounced in the following order: EGC, GC > EGCG > T > ECG > C, EC > CG. The tyrosinase inhibition was more pronounced for GC, EGC, EGCG, ECG, CG and T. However, these initial response intensities need to be underlined by further detailed quantitative studies. The effects observed against Gram-negative *A. fischeri* bacteria and β-glucuronidase were assigned to other components in the tea extracts due to the quite different bioactivity pattern across all samples and the comparatively very low responses for the 11-bioactive-compound mixture.

All in all, the diverse bioactivity responses observed for the commercially available powdered tea extracts were in accordance to their specifications. For higher specified contents of catechins or theaflavins, higher bioactivity responses were obtained. The tea extract ID 4 with the highest specified content of 60% theaflavins, showed the highest bioactivity responses in the theaflavins’ range below *hR*_F_ 30. Any catechins or flavonols were not specified. Both latter were not or hardly detected, as evident in the physicochemical and effect-directed profiling for the tea extract ID 4. As another proof, tea extract IDs 5 and 11 with a specified lower, respective 20% and 15%, polyphenol content showed a less intense functional bioactivity pattern, if compared to IDs 8, 9, and 13 with respective specified polyphenol contents of 95%, 98% and 90%. As expected, the bioactivity responses were much higher in the powdered tea extracts than in the extracts of the reference leaves of green, white and black tea (IDs 1, 10 and 20, respectively, Appendix A). This comparative side by side visualization makes clear the added value of such powdered tea extracts when integrated in functional food products. A picture says more than a thousand words. Such an evaluation of proposed activities of individual ingredients in functional food products can be transferred to other botanicals. However, the decisive advantage of such a multi-imaging and effect-directed profiling of hundreds of samples is the non-targeted detection of any bioactive compound, which can also be a contaminant or residue or adulteration that may affect the health of consumers. Such information gets more and more important and can be obtained by direct elution of the bioactive zone of interest into a high-resolution mass spectrometry system for further characterization.

## 4. Conclusions

An effect-directed profiling, exploiting multi-imaging (UV/Vis/FLD) and eight different biological and enzymatic assays, was successfully developed and demonstrated to provide health-related information on functional powdered tea extracts on the market. The developed bioactivity profiling is also able to discover unknown bioactive compounds. This asset is very important for the global chain of food production and processing. The HPTLC–EDA–UV/Vis/FLD/bioluminescence profiling was shown to ensure the authenticity based on the non-volatile part (characteristic profile) and involved effects deriving from individual ingredients (bioactivity profiles) on the one hand, and to discover product defects and undesirable components (UV/Vis/FLD/EDA multi-imaging profiles) on the other hand. The comprehensive information obtained by eight biological and enzymatic assays contribute to new insights in the increasing variety of tea extracts on the global market. As comminution to a powder makes the visual proof impossible, this newly developed profiling can be used for product control to examine the product quality of tea extract powders. Such a profiling is generally proposed for quality and safety control of functional food and health food.

## Figures and Tables

**Figure 1 antioxidants-10-00117-f001:**
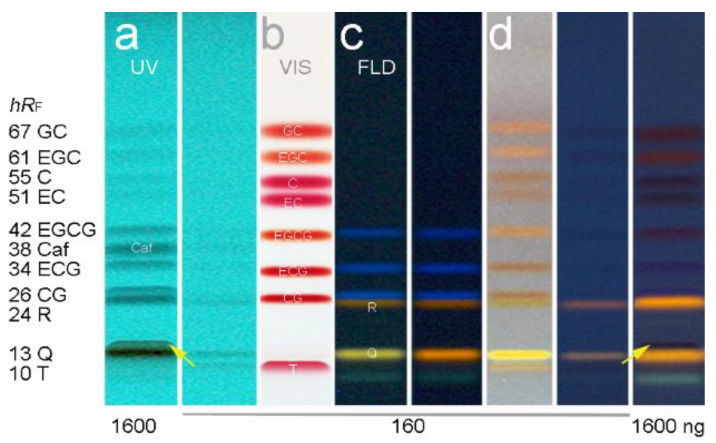
HPTLC–UV/Vis/FLD chromatograms of the developed separation of the 11-bioactive-compound mixture (0.16 and 1.6 µg/band) on the 10 × 10 cm HPTLC plate silica gel 60 RP–18 WF_254_s using acetonitrile–water–citric acid (1.8 mL + 6 mL + 23 mg) at UV 254 nm (**a**), at Vis after derivatization with the Fast Blue B salt reagent (**b**), at FLD via the natural product reagent (**c**), Q turned orange by PEG 400, but of no advantage), and after the reagent sequence of first b then c (**d**); the arrow marks the *hR*_F_ shift of T for higher T amounts (from below to top of Q, in detail in Appendix A).

**Figure 2 antioxidants-10-00117-f002:**
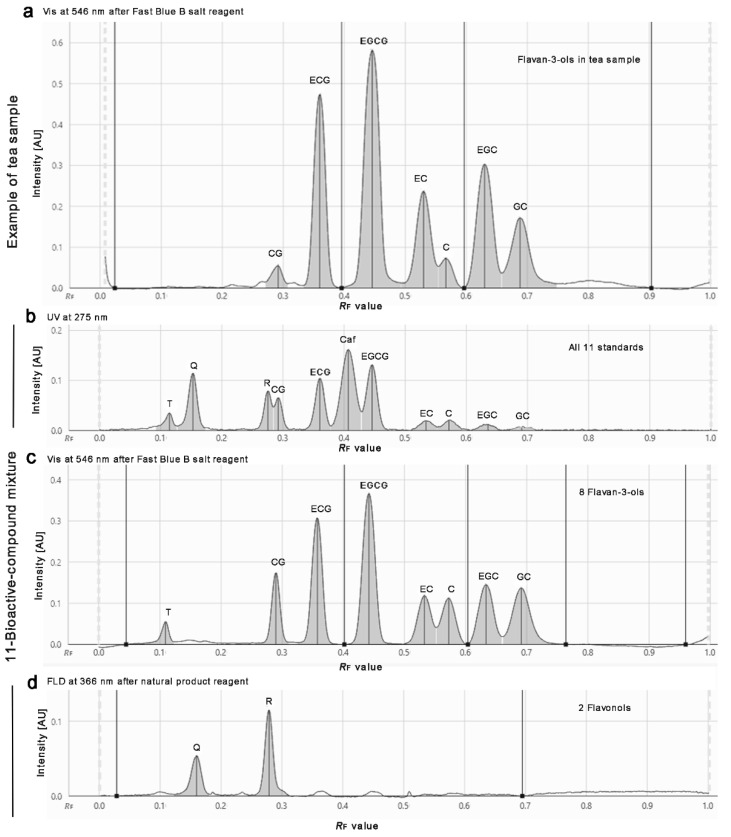
Example densitograms of a commercially available powdered tea extract product (**a**) and the 11-bioactive-compound mixture (**b**–**d**) separated (as Figure 1) on the same plate: the densitometric scan was performed at 275 nm for UV absorbance measurement of all 11 compounds, but especially caffeine (**b**), after selective derivatizations (reagent sequence) first with the Fast Blue B salt reagent at 546 nm for Vis absorbance measurement of all flavan-3-ols (**a**,**c**), and then natural product reagent at 366/>400 nm for FLD measurement of all flavonols (**d**).

**Figure 3 antioxidants-10-00117-f003:**
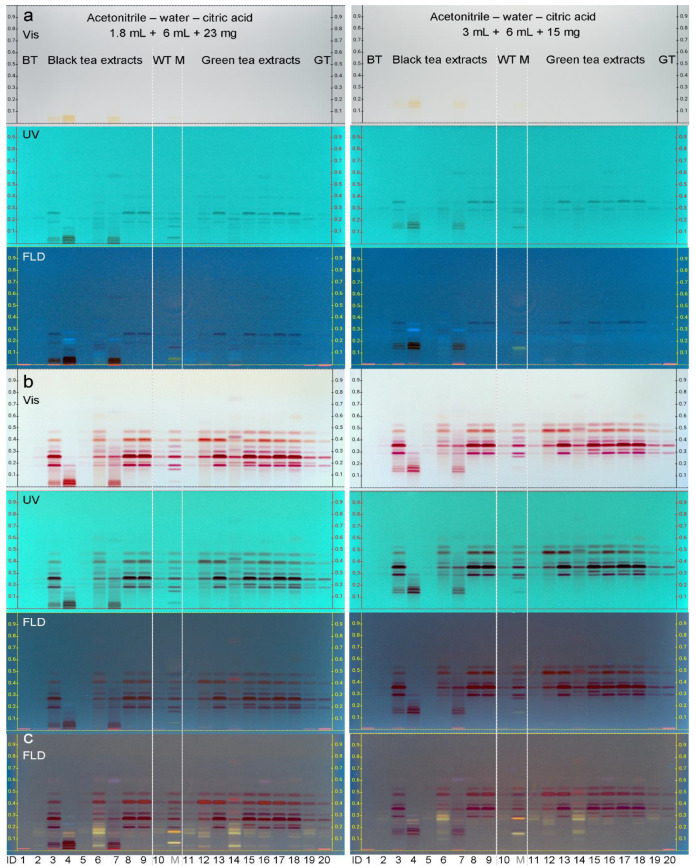
Physicochemical profiling: HPTLC–UV/Vis/FLD chromatograms of the 20 tea extracts (0.8 µL, 8 µg/band; black/white/green tea: BT/WT/GT; IDs in Appendix A) and the 11-bioactive-compound mixture (M, 0.075–0.675 µg/band, depending on the individual compound amount, which was adjusted to the findings in the tea samples) separated on LiChrospher^®^ HPTLC plate silica gel 60 RP–18 WF_254_s using acetonitrile–water–citric acid (1.8 mL + 6 mL + 23 mg) versus (3 mL + 6 mL + 15 mg) before (**a**) and after reagent sequence of first Fast Blue B salt reagent (**b**); detection of 8 purple-red flavan-3-ols) and then natural product reagent (**c**); detection of yellow/orange fluorescent flavonols).

**Figure 4 antioxidants-10-00117-f004:**
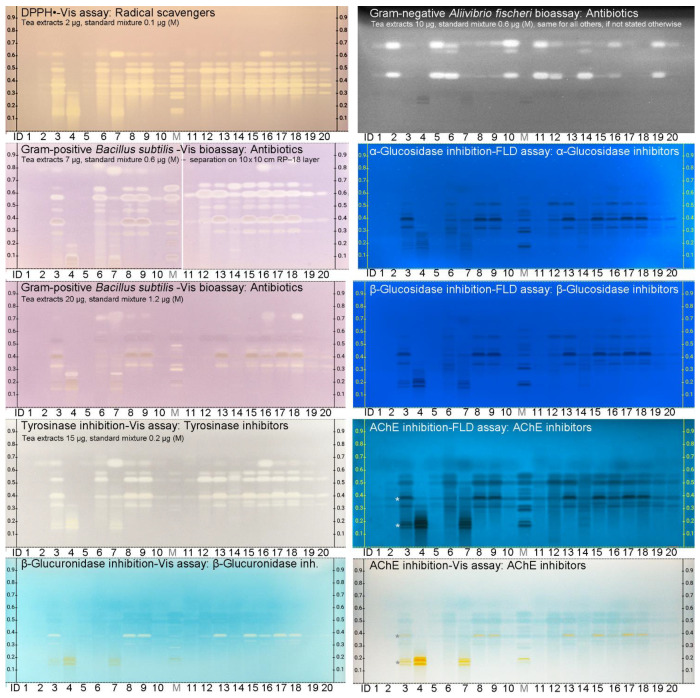
Effect-directed profiling: HPTLC–EDA–UV/Vis/FLD autograms (*A. fischeri* bioluminescence after 45 min depicted as greyscale image) of the 20 tea sample extracts (1.0 µL, 10 µg/band, if not stated otherwise; IDs in Appendix A) and the 11-bioactive compound mixture (M, 0.6 µg/band, if not stated otherwise) on LiChrospher^®^ HPTLC plate silica gel 60 RP–18 WF_254_s using acetonitrile–water–citric acid (3 mL + 6 mL + 15 mg). The *B. subtilis* bioassay did hardly work on the spherical layer, and thus, was performed on the 10 cm × 10 cm HPTLC plate silica gel 60 RP–18 WF_254_s using acetonitrile–water–citric acid (1.8 mL + 6 mL + 23 mg). Using indoxyl acetate as substrate, the AChE inhibition–Vis assay converted T-like representatives into yellow zones (marked*).

## Data Availability

The data presented in this study are available on request from the corresponding author.

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
