# Peer review of "Effect-Directed Profiling of Powdered Tea Extracts for Catechins, Theaflavins, Flavonols and Caffeine"

_antioxidants, 2021, doi:10.3390/antiox10010117_

Round 1

Reviewer 1 Report

Comments to Author:

Ms. Ref. No.: ID: antioxidants-1040294

Title: Effect-Directed Profiling of Tea and Tea Extracts For Catechins, Theaflavins, Flavonols and Caffeine

Overview and general recommendation:

I recommend that a minor revision is warranted. I explain my concerns in more detail below. I ask that the authors specifically address each of my comments in their response.

  1. This study aims as an objective to develop a fast effect-directed, multi-imaging HPTLC profiling for tea and tea extracts. The current study is on a topic of relevance and general interest to the readers of the journal with a relative impact on scientific research.
  2. In general, a clear thread should be woven through the text linking the introduction, methods, results, and conclusions. After carefully checking the article, I would suggest the following improvements to the article.
  • Keywords: Please provide the relevant keywords (g., HTPLC is the same with high-performance thin-layer chromatography). The authors must use only or abbreviation or current name of analytical technique! Maybe “reversed-phase high-performance thin-layer chromatography” can be the right word! Other suggestions for keywords can be Camelia sinensis and extract as well! Please reformulate!
  • In Conclusion Section clarify the rationale and reasons for the used methods and the interest to the readers! Explain how other readers can use this work in the future!
  1. Raise the quality of English in the manuscript!

More specifically

  1. Lines 61-63: Please provide the abbreviation for all analytical techniques (e.g., HPLC, HPTLC, GC).
  2. Line 118: our site, please remove!
  3. Line 131: 6 HPTLC replace with six HPTLC. Please check in the whole MS!

Author Response

This reviewer report 1 is identical to the Academic Editor Notes, which I have just responded.

Reviewer 2 Report

This manuscript is interesting enough that the authors used a multi-imaging analysis for the major bioactive compounds in tea. The development of a bioactive compound analysis method in tea using HPTLC is thought to be helpful in developing the field of HPTLC analysis. In particular, the derivatization part is thought to be significant in excluding the matrix of food samples when analyzing target compounds such as catechins and theaflavins.

However, I think there is two major problems for this manuscript to be published.

First, this manuscript is more like a report than a research paper. The reasons are as follows.

1). The authors focus only on the process and method of obtaining the results.

2). The presentation of the theoretical background for the experiment and the theoretical interpretation for the results are very scarce.

3). The authors have described the text without clear evidence, such as …known to be bioactive in tea were selected (Line 220), bioactivity responses were expected (Line 223), 11 bioactive compounds have not been shown to be separated at one go in the reported methods (Line 237), any eficiency, falsification or adulteration would also be detectable (Line 323).

Having only 5 references in the Results and Discussion section reflects these indirectly.

Detailed comments on 3).

(Line 220): Selected evidence required

(Line 223): Results of previous studies or experiments on bioactive compounds required

(Line 237): Reported in a paper published in Antioxidants.

(Line 323): Requires results of detected falsification or adulteration.

Next, In my opinion, this paper is far from antioxidants's scope

This manuscript suggested potential antioxidants in tea, but did not provide a quantitative assessment. In addition, the target compounds evaluated in this study have already been quantitatively evaluated in many other studies. Therefore, since there is no quantitative evaluation of potential antioxidants present in tea and evaluation of their antioxidant activity. I think that the evaluation using HPTLC is suitable for confirming the presence of target components, but difficult quantification is a fatal disadvantage. In order to overcome these shortcomings, I think it is necessary to explain the novelty, necessity, and merits of HPTLC analysis, and the results (Quantification table of samples) accordingly.

Other minors,

It is necessary to confirm that both authors are required to provide their email addresses.

You need to avoid repetition and select key keywords. (bioassay and biological assay, HPTLC and

high-performance thin-layer chromatography…).

The abbreviation definition should be reviewed. (HPLC, TLC, RP).

The purity notation for Quercetin and Caffeine should be unified as for other compounds.

You need to correct the multiplication expression (such as Line 116). Line 172 is considered as the standard. Line 127 (17000 x g ) is wrong.

Plus sings must be spaced.

Volume symbols are thought to be more common in lowercase italics (v/v).

The chiral expression (D/L) should have a relatively small font size.

Range sign such as 15-20 min must be distinguished from the minus sign.

The first half of the conclusion is thought to be irrelevant to the results obtained in the paper.

What ‘insights’ (Line 387) can be obtained based on the results of this study should be described in the discussion section or in the preceding sentence.

In conclusion, my opinion is that this version of manuscript is more suitable for method development journals than antioxidants.

Author Response

This manuscript is interesting enough that the authors used a multi-imaging analysis for the major bioactive compounds in tea. The development of a bioactive compound analysis method in tea using HPTLC is thought to be helpful in developing the field of HPTLC analysis. In particular, the derivatization part is thought to be significant in excluding the matrix of food samples when analyzing target compounds such as catechins and theaflavins.

However, I think there is two major problems for this manuscript to be published.

First, this manuscript is more like a report than a research paper. The reasons are as follows.

1). The authors focus only on the process and method of obtaining the results.

=> The research was focused on the development of a bioanalytical method suited for profiling of powdered tea extracts. Apart from the antioxidative power, further effects were revealed in these products. This method can be used in the future to examine the product quality of powders, which is not easy to judge visually. We also added this in the Conclusions part.

2). The presentation of the theoretical background for the experiment and the theoretical interpretation for the results are very scarce.

=> We added more information in the Results and Discussion part. For the theoretical background, we found the information given to be sufficient to reproduce the experiments. With regard to the assay experiments, we referred to existing literature, if available.

3). The authors have described the text without clear evidence, such as …known to be bioactive in tea were selected (Line 220), bioactivity responses were expected (Line 223), 11 bioactive compounds have not been shown to be separated at one go in the reported methods (Line 237), any eficiency, falsification or adulteration would also be detectable (Line 323).

=> See below responded in detail.

Having only 5 references in the Results and Discussion section reflects these indirectly.

=> We added further references in the Results and Discussion section.

Detailed comments on 3).

(Line 220): Selected evidence required

(Line 223): Results of previous studies or experiments on bioactive compounds required

=> L220/223: We added a reference, in which it is mentioned that these compounds are known to be bioactive in tea.

(Line 237): Reported in a paper published in Antioxidants.

=> L237: We added references, in which it is evident that not 11 compounds have been separated in reported HPTLC literature so far.

(Line 323): Requires results of detected falsification or adulteration.

=> L323: We added “owed to the different detectors used” to make it more clear.

Next, In my opinion, this paper is far from antioxidants's scope

This manuscript suggested potential antioxidants in tea, but did not provide a quantitative assessment. In addition, the target compounds evaluated in this study have already been quantitatively evaluated in many other studies. Therefore, since there is no quantitative evaluation of potential antioxidants present in tea and evaluation of their antioxidant activity. I think that the evaluation using HPTLC is suitable for confirming the presence of target components, but difficult quantification is a fatal disadvantage. In order to overcome these shortcomings, I think it is necessary to explain the novelty, necessity, and merits of HPTLC analysis, and the results (Quantification table of samples) accordingly.

Other minors,

It is necessary to confirm that both authors are required to provide their email addresses.

=> We confirm that both emails were typed in and listed in the submission platform.

You need to avoid repetition and select key keywords. (bioassay and biological assay, HPTLC andhigh-performance thin-layer chromatography…).

=> We changed some of the keywords based on the recommendations.

The abbreviation definition should be reviewed. (HPLC, TLC, RP).

=> We use abbreviations only, if the word is re-used. We checked this upon consistency.

The purity notation for Quercetin and Caffeine should be unified as for other compounds.

=> We added this missing information.

You need to correct the multiplication expression (such as Line 116). Line 172 is considered as the standard. Line 127 (17000 x g ) is wrong.

=> We corrected the multiplication expression. Note that it need to be written in italics in L127.

Plus sings must be spaced.

=> Done.

Volume symbols are thought to be more common in lowercase italics (v/v).

=> We prefer the official IUPAC writing (capital in italics).

The chiral expression (D/L) should have a relatively small font size.

 => Done. We used the font size 9 (instead of 10).

Range sign such as 15-20 min must be distinguished from the minus sign.

=> Done.

The first half of the conclusion is thought to be irrelevant to the results obtained in the paper.

=> We revised the first part of the Conclusions section.

What ‘insights’ (Line 387) can be obtained based on the results of this study should be described in the discussion section or in the preceding sentence.

=> We rephrased it and made the outcome more clear in the Results and Discussion section.

In conclusion, my opinion is that this version of manuscript is more suitable for method development journals than antioxidants.

=> We think that also a methodical paper contributes to the science on antioxidants, as analytical chemistry is the basis of all knowledge.

Reviewer 3 Report

The manuscript is very interesting because the number of botanical powders specified are rich in healthy compounds, and the paper focused on the need of new bioanalytical tools, capable for a cost-efficient screening and bioactivity information. The paper developed an effect-directed profiling and demonstrated to provide health-related information on functional tea extracts on the market. The developed bioactivity profiling was also capable of  discovering unknown bioactive compounds, which is very important for the given global chain of food production and processing. The informations obtained by seven biological and enzymatic assays contribute to new insights in the increasing variety of tea extracts on the global market.

Author Response

=> Thank you for this reflection.

Round 2

Reviewer 2 Report

This manuscript has improved a lot.

Nevertheless, I still have two questions.

1). The authors answered that manuscript is the contributes to the science on antioxidants, as analytical chemistry is the basis of all knowledge. But I disagree with this.

I think one of the reasons why so many disciplines are divided is to increase reader access and improve expertise in the field.

2). This manuscript profiled the major polyphenols in green tea with HPTLC. However, the major polyphenols of green tea have already been sufficiently profiled with tandem MS or TOF.

Therefore, I don't think this study has novelty unless there is sufficient evidence to use HPTLC.

In addition, in order for the profiling method developed with HPTLC to have academic value, it must be supported by a validation process.

This study has been well written analytically and chemically. Also, I respect the authors' research process and results.

However, I think this manuscript is more suitable for fields such as separations, J. separation sci., analytical methods, food chem., J. Chromatography-B, and analytical chemistry than antioxidants.

Miners:

55-59: A “personal communication” basis for “the current control of powdered tea extract products needs to be reflected” is not suitable for scientific soundness.

83: Theaflavins that appear suddenly have no context and are unnatural.

Reference 43 appears to be less relevant.

287-296: Not only is the introduction and content overlapping, it is a too much information. It needs to be summarized in one or two lines.

Change Figure 2. The picture is intertwined.

Check out italics in reference no. 21.

Reference 45 cannot be found in the text.

I am concerned that references 43-47 will be self-cited.